# Mycobacterial infection-induced miR-206 inhibits protective neutrophil recruitment via the CXCL12/CXCR4 signalling axis

Kathryn Wright[1,2], Kumudika de Silva[2], Karren M. Plain[2], Auriol C. Purdie[2], Tamika A. Blair[3], Iain G. Duggin[3], Warwick J. Britton[1,4], Stefan H. Oehlers[1,5]*

1 Tuberculosis Research Program at the Centenary Institute, The University of Sydney, Camperdown, New South Wales, Australia, 2 The University of Sydney, Faculty of Science, Sydney School of Veterinary Science, Sydney, New South Wales, Australia, 3 ithree Institute, University of Technology Sydney, Ultimo, New South Wales, Australia, 4 Department of Clinical Immunology, Royal Prince Alfred Hospital, Camperdown, New South Wales, Australia, 5 The University of Sydney, Faculty of Medicine and Health & Marie Bashir Institute, Camperdown, New South Wales, Australia

* stefan.oehlers@sydney.edu.au

**Data Availability Statement:** All relevant data are within the manuscript and its Supporting Information files.

## Abstract

Pathogenic mycobacteria actively dysregulate protective host immune signalling pathways during infection to drive the formation of permissive granuloma microenvironments. Dynamic regulation of host microRNA (miRNA) expression is a conserved feature of mycobacterial infections across host-pathogen pairings. Here we examine the role of miR-206 in the zebrafish model of *Mycobacterium marinum* infection, which allows investigation of the early stages of granuloma formation. We find miR-206 is upregulated following infection by pathogenic *M. marinum* and that antagomir-mediated knockdown of miR-206 is protective against infection. We observed striking upregulation of *cxcl12a* and *cxcr4b* in infected miR-206 knockdown zebrafish embryos and live imaging revealed enhanced recruitment of neutrophils to sites of infection. We used CRISPR/Cas9-mediated knockdown of *cxcl12a* and *cxcr4b* expression and AMD3100 inhibition of Cxcr4 to show that the enhanced neutrophil response and reduced bacterial burden caused by miR-206 knockdown was dependent on the Cxcl12/Cxcr4 signalling axis. Together, our data illustrate a pathway through which pathogenic mycobacteria induce host miR-206 expression to suppress Cxcl12/Cxcr4 signalling and prevent protective neutrophil recruitment to granulomas.

## Author summary

Mycobacterial infections cause significant disease burden to humans and animals, the most widely known example being tuberculosis which has killed more humans than any other infectious disease throughout history. Infectious mycobacteria are highly evolved to hijack host processes, including the very immune cells tasked with destroying them. MicroRNAs are host molecules that control wide-ranging programs of host gene expression and are important in the immune response to infections. Here we use the zebrafish model of mycobacterial infection to determine the role of the infection-induced

**Funding:** Funding for this study was provided to SHO: Australian National Health and Medical Research Council Grant [APP1099912] https://www.nhmrc.gov.au/, The University of Sydney Fellowship [G197581] https://www.sydney.edu.au/, NSW Ministry of Health under the NSW Health Early-Mid Career Fellowships Scheme [H18/31086] https://www.health.nsw.gov.au/; to KdS, KMP, ACP: Meat and Livestock Australia Grant [P. PSH.0813] https://www.mla.com.au/; to KW Meat and Livestock Australia Grant Higher Degree Research scholarship to KW [P.PSH.0813 HDR scholarship] https://www.mla.com.au/; to WJB: Australian National Health and Medical Research Council Centres of Research Excellence Grant [APP1153493] https://www.nhmrc.gov.au/. The funders had no role in study design, data collection and analysis, decision to publish, or preparation of the manuscript.

**Competing interests:** The authors have declared that no competing interests exist.

microRNA miR-206 in the host response to infection. We found pathogenic mycobacteria trigger the host to produce more miR-206 in order to suppress the otherwise protective recruitment of neutrophils to sites of infection via the host Cxcl12/Cxcr4 signalling pathway. Our study provides new insight into the role of mycobacterial infection-induced miR-206 function in the context of a whole host.

## Introduction

Pathogenic mycobacteria, including the causative agents of tuberculosis and leprosy, are capable of appropriating host signalling and immune pathways to increase their survival and establish chronic infection in cell-rich granulomas, which support mycobacterial growth and latent survival [1,2]. The interaction between mycobacteria and host immune cells is therefore key to the effective prevention of infection or an ineffective immune response and bacterial persistence.

MicroRNA (miRNA) are short, non-coding RNA of approximately 22 nucleotides that can post-transcriptionally regulate gene expression and transcript abundance through gene silencing. miRNA bind to the untranslated region (UTR) of mRNA to regulate the stability of target genes through degradation or suppression, reducing protein translation [3]. Expression of miRNA is dynamically regulated in mycobacterial infection, suggesting a key role in the host response to infection by the modulation of downstream genes and protein expression [4–6].

The expression of miRNA is notably altered in mycobacterial infections and have been suggested as biomarkers for several conditions [7,8]. Regulation of miRNA by invading mycobacteria can further modulate the host immune responses, altering the survival and persistence of bacteria [5,9]. Further, the potential of miRNA to not only identify infected and exposed individuals, but also as prognostic markers and markers of treatment, has resulted in greater research into their expression during infection [10]. It is therefore key to identify differentially expressed miRNA and explore their mechanism of function, and the outcome of their modulation by mycobacteria.

miR-206 is a member of the muscle-associated myomiR family and is characteristically associated with myoblast differentiation and muscle development [11–14]. However, miR-206 has been recently found to be differentially regulated in mycobacterial infection of THP-1 monocyte-like cells [15]. Infection of THP-1 cells with *Mycobacterium tuberculosis* revealed a role for miR-206 in the regulation of proinflammatory cytokine responses through reducing TIMP3 expression [15]. miR-206 has also been implicated in viral pathogenesis, reducing replication of influenza virus, and in neuroinflammation [16,17]. While miR-206 clearly has a diverse range of biological functions, the *in vivo* role of miR-206 during mycobacterial infection remains undetermined.

Here we use the zebrafish-*Mycobacterium marinum* model to investigate the *in vivo* function of dre-miR-206-3p (miR-206) in mycobacterial infection, and the impact on host gene expression and neutrophil recruitment. Zebrafish (*Danio rerio*) present an ideal model to study host-pathogen interactions due to their optical transparency, easy genetic manipulation, and the availability of fluorescent transgenic reporter lines. During early embryogenesis, zebrafish embryos possess a functional immune system, with immune cells such as macrophages and neutrophils available and actively responding to invading pathogens [18–20]. A natural pathogen of zebrafish, *M. marinum*, is a close relative of *M. tuberculosis*, and closely mimics pathogenesis and pathology of virulent mycobacterial infections [21,22]. Early infection with *M. marinum* results in a widespread systemic infection, while chronic infection is established

by approximately three to five days post infection with the formation of granulomas. The ability to visualise the interaction between host immune cells and bacteria emphasises the applicability of the zebrafish-*M. marinum* model of mycobacterial pathogenesis.

Zebrafish are also an established model for the investigation of the outcome of host-mycobacteria interactions, and to evaluate host gene function, including the role of chemokine signalling [23–25]. Regulation of neutrophil based inflammation and motility by miRNA has also been investigated using a zebrafish model [26–30], highlighting the applicability of the zebrafish model to study conserved miRNA functions within host-pathogen interactions.

## Results

### Zebrafish miR-206 expression is responsive to *M. marinum* infection

To determine if miR-206 is responsive to *M. marinum* infection in zebrafish, embryos were infected with *M. marinum* via caudal vein injection and miR-206 expression was measured by quantitative (q)PCR at 1, 3, and 5 days post infection (dpi) (S1 Data). Infection with *M. marinum* increased miR-206 expression at 1 and 3 dpi, but decreased miR-206 expression at 5 dpi compared to uninfected controls (Fig 1A).

### AntagomiR abrogates infection-induced miR-206 expression

To determine the efficacy of antagomiR-mediated miRNA knockdown during *M. marinum* infection, embryos were injected with miR-206 antagomiR at the single-cell stage and infected at 1.5 days post fertilisation (dpf). miR-206 expression levels were analysed at 1-, 3-, and 5- days post infection (dpi). At 1 and 3 dpi, miR-206 was increased in *M. marinum* infected embryos compared to control uninfected (p = 0.027 and p = 0.039 respectively). AntagomiR knockdown effectively reduced the miR-206 level relative to infected embryos at both timepoints, demonstrating effective knockdown of infection-induced miR-206 expression by antagomiRs (Fig 1B).

By 5 dpi, the efficacy of the antagomiR was reduced and there was no difference in miR-206 expression between treatments which precluded analysis of later timepoints (S1 Fig).

### miR-206 knockdown reduces *M. marinum* burden

As miR-206 was modulated during infection, its effect on disease was assessed through assessing the bacterial burden following antagomiR knockdown. There was no difference in bacterial burdens between miR-206 knockdown and control embryos at 1 dpi, however by 3 dpi, knockdown embryos had a significantly lower burden than control embryos. (Fig 1C).

### Infection-induced miR-206 upregulation is driven by mycobacterial virulence factors

To investigate whether the decreased bacterial burden in miR-206 knockdown embryos was a general response to foreign pathogens or a more directed response, embryos were infected with either ΔESX1 *M. marinum* or uropathogenic *Escherichia coli* (UPEC). ΔESX1 *M. marinum* lack the key type VII secretion system and are far less virulent as they are unable to lyse host cell membranes to escape the phagosome [31]. In comparison to mycobacteria, UPEC cause an acute sepsis infection and are an example an extracellular bacterium.

Expression of miR-206 was analysed by qPCR in embryos infected with WT *M. marinum*, ΔESX1 *M. marinum*, or UPEC at 1 dpi (Fig 1D). Infection with ΔESX1 *M. marinum* increased miR-206 expression, however this response was less than the level induced by infection with virulent WT *M. marinum*. Conversely, miR-206 was decreased in embryos infected with UPEC.

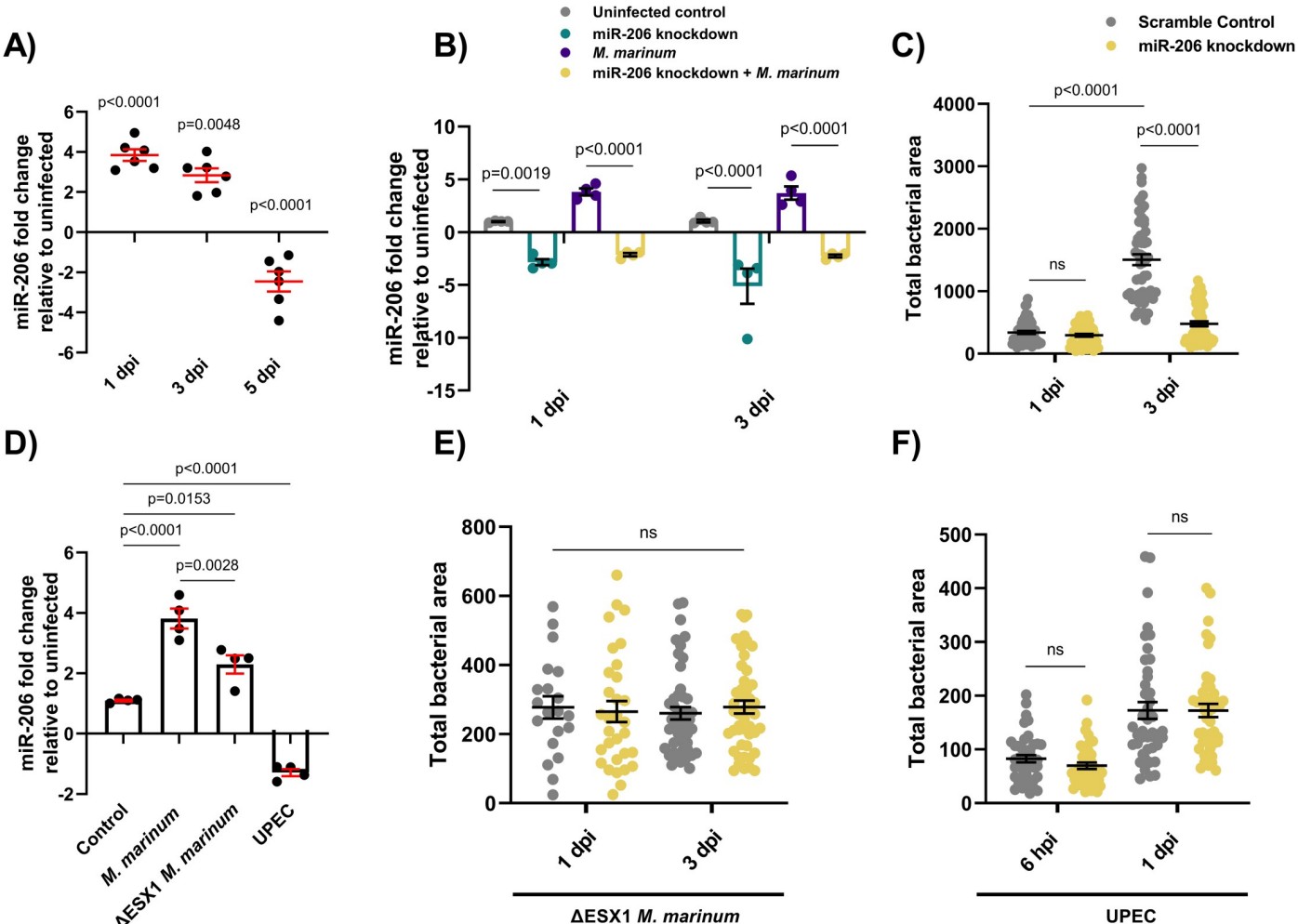

**Fig 1. Infection-induced miR-206 expression alters bacterial burden.** (A) Expression of miR-206 analysed by qPCR at 1, 3, and 5 dpi. (B) Expression of miR-206 in uninfected and infected antagomir-injected embryos (miR-206 knockdown). (C) *M. marinum* burden in miR-206 knockdown embryos at 1 and 3 dpi. (D) Expression of miR-206 at 1 dpi following infection with either wild-type (WT) *M. marinum*, ΔESX1 *M. marinum*, or UPEC. (E) ΔESX1 *M. marinum* burden in miR-206 knockdown embryos at 1 and 3 dpi. (F) UPEC burden in antagomir-injected embryos at 6 hpi and 1 dpi. Each data point represents a single measurement, with the mean and SEM shown. For qPCR analysis, each data point represents 10 embryos, and contains 2 biological replicates. Bacterial burden analysis data points (WT *M. marinum*, ΔESX1 *M. marinum*, and UPEC) represent individual embryos (n = 40–50 embryos per group) and are representative of 2 biological replicates.

ΔESX1 *M. marinum* infection burdens were unaffected by miR-206 knockdown at either 1 or 3 dpi (Fig 1E). Similarly, there was also no difference in UPEC burden levels in miR-206 knockdown embryos at either 6 hours post infection (hpi) or 1 dpi despite an increase in infection between timepoints (Fig 1F). These results indicate that the impact on bacterial burden in miR-206 knockdown embryos is driven by *M. marinum* virulence factors.

## miR-206 target mRNA gene expression patterns are conserved during *M. marinum* infection of zebrafish

To further investigate the functional relevance of miR-206 in mycobacterial infection, a list of potential mRNA target genes was compiled through published experimentally observed targets and bioinformatic target prediction algorithms [32–35] (S1 Table).

Expression of selected potential target genes of miR-206 was analysed by qPCR at 2 dpf, with increased expression in knockdown samples expected to indicate targeting by miR-206

(Fig 2A). From the expression profiling data and prediction analysis, *cxcl12a* and *cxcr4b* were considered to be likely targets of miR-206 (S2 Fig). Expression of these genes was increased by *M. marinum* infection and in both knockdown treatments, suggesting they may be active during infection and contributing to the decreased bacterial burden observed in miR-206 knockdown embryos. These genes were also of particular interest as the Cxcl12/Cxcr4 pathway has been previously implicated in zebrafish immunity (14, 15).

The expression of *cxcr4b* and *cxcl12a* was analysed at 1, 3, and 5 dpi in *M. marinum* WT embryos (S3 Fig). Expression of these genes was increased throughout infection, with the exception of *cxcl12a* at 5 dpi, and did not appear to be dramatically increased in response to the reduced amount of miR-206 at 5 dpi.

Expression of *cxcr4a*, *cxcr4b*, *cxcl12a* was analysed at 3 dpi, showing that knockdown of miR-206 significantly increased the transcript abundance of *cxcr4b* and *cxcl12a* in infected embryos compared to *M. marinum* infection alone and uninfected controls (Fig 2B–2D).

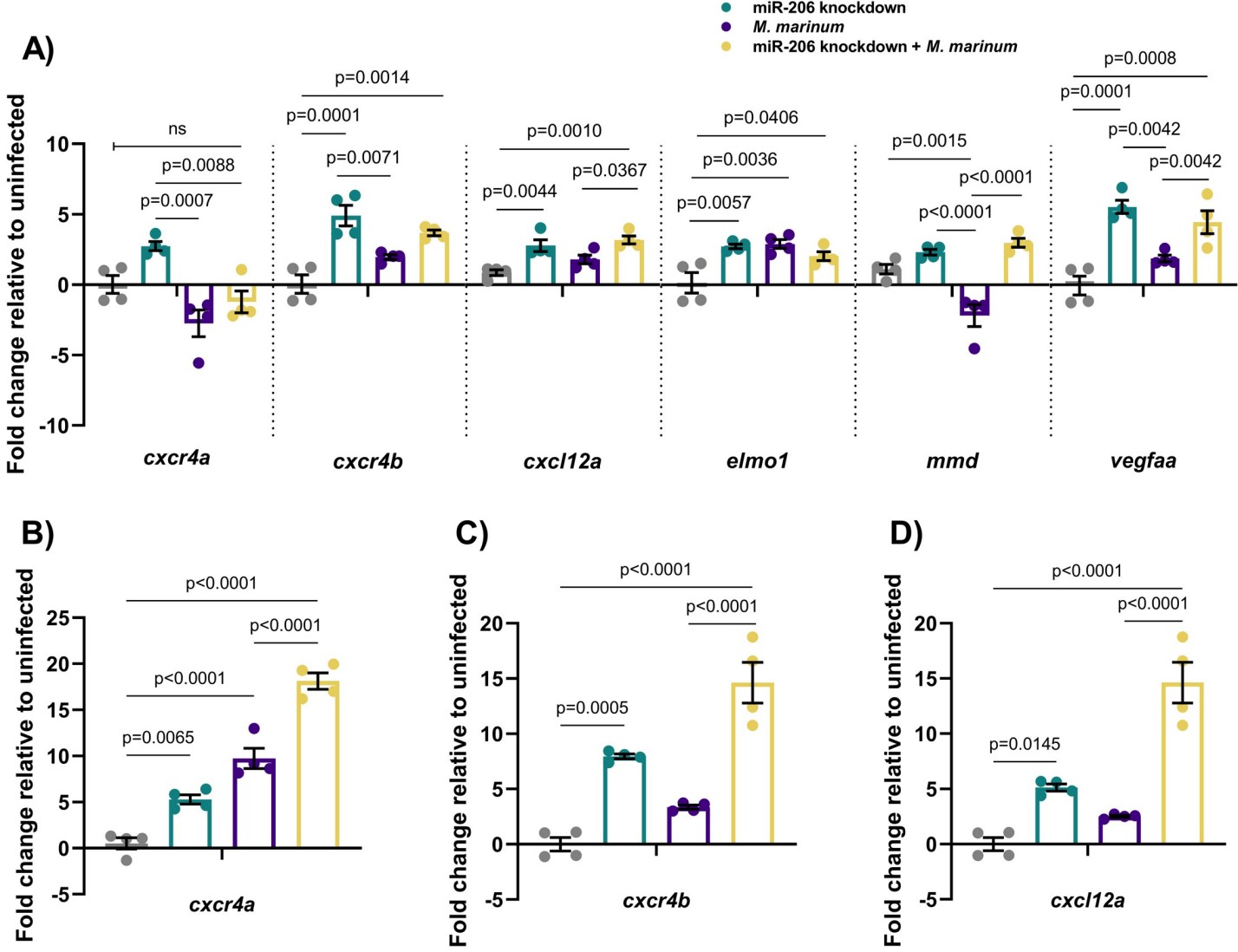

**Fig 2. Expression profiles of potential mRNA targets of miR-206.** (A) Expression of candidate target genes measured by qPCR at 1 dpi in miR-206 knockdown embryos. (B-D) Expression of zebrafish CXCL12/CXCR4 pathway ortholog genes at 3 dpi. Each data point represents a single measurement of 10 pooled embryos and 2 biological replicates, with the mean and SEM shown.

## Knockdown of miR-206 increases neutrophil recruitment to, and retention at, sites of *M. marinum* infection

As miR-206 knockdown increased the expression of *cxcl12a* and *cxcr4* genes, which are involved in neutrophil migration and retention of cells at sites of infection and inflammation [28,36–38], the neutrophil response to *M. marinum* infection in miR-206 knockdown treated embryos was assessed by live imaging of transgenic *Tg(lyzC:GFP)^{nz117}* or *Tg(lyzC:DsRed2)^{nz50}* embryos, where neutrophils are fluorescently labelled.

First, static imaging was performed at 1 and 3 dpi to measure total neutrophil numbers in infected embryos. At each timepoint, miR-206 knockdown embryos had a significantly higher total number of neutrophils compared to control (Fig 3A and 3B). We noted strong overlap of neutrophils with *M. marinum* but the location of granulomas around the caudal haematopoietic tissue confounded quantification of granuloma-associated neutrophils. Total neutrophils were also analysed at 5 dpi, however as antagomiR effectiveness decreases past 3 dpi, no difference was observed between control and miR-206 knockdown embryos (S4 Fig).

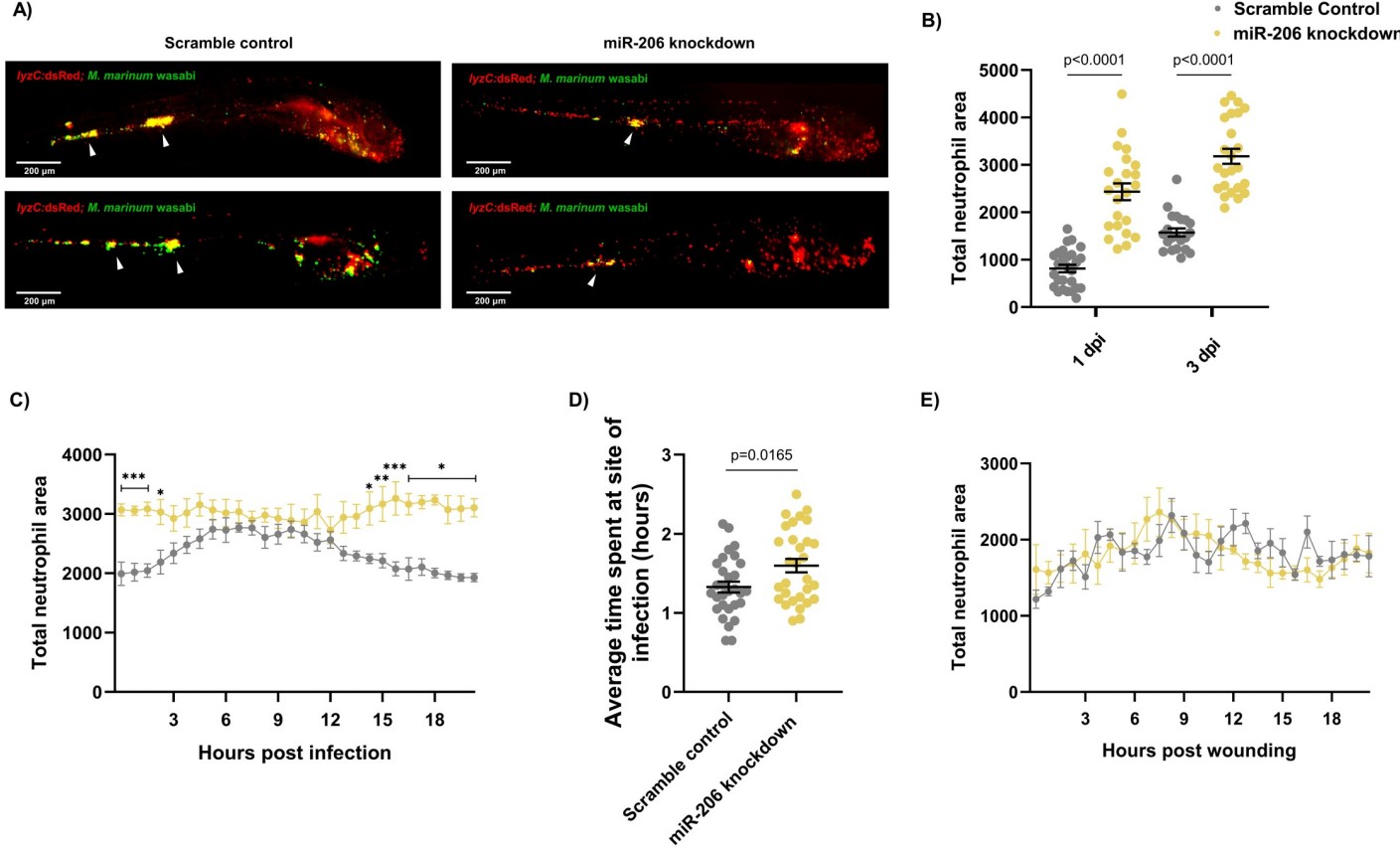

**Fig 3. Infection-induced miR-206 expression alters the host neutrophil response.** (A) Representative images of infection phenotype at 3 dpi in control and miR-206 knockdown embryos. White arrows indicate bacterial foci. Neutrophils are red (*lyzC:dsred*) and *M. marinum* is green (wasabi); co-localisation is indicated by yellow fluorescence. (B) Measurement of whole-body neutrophil fluorescent area at 1 and 3 dpi in miR-206 knockdown embryos. (C) Measurement of neutrophil levels following trunk infection with *M. marinum* in miR-206 knockdown embryos. (D) Measurement of neutrophil retention at sites of infection following trunk infection with *M. marinum* in control and miR-206 knockdown embryos. (E) Measurement of neutrophil recruitment to a sterile tail fin wound in miR-206 knockdown embryos. Each data point represents a single measurement, with the mean and SEM shown. For time-lapse imaging, each data point represents the mean of 6 foci of infection from 6 separate embryos. Neutrophil retention was measured by selecting 10 random cells in 3 embryos per group and measuring the time spent at the granuloma. Bacterial burden analysis was performed on 15–20 embryos per treatment. Graphs are representative of 2 biological replicates. * P < 0.05, ** p < 0.01, *** p < 0.001.

To determine if miR-206 knockdown increased the number of infection-associated neutrophils, embryos were injected with *M. marinum* into the trunk (away from the caudal haematopoietic tissue) and subjected to time-lapse imaging [39]. Knockdown embryos had significantly more neutrophils at the site of infection for the first 2.5 hours of infection compared to control infected embryos (Fig 3C). While neutrophil migration in control infected embryos began to wane at approximately 12 hpi (S1 Movie), the response in the knockdown embryos was sustained and higher numbers of neutrophils were maintained at the site of infection (S2 Movie).

To examine if the increased mobilisation of neutrophils in the *M. marinum*-infected miR-206 knockdown embryos was dependent on mycobacterial infection cues or an intrinsic feature of neutrophils in miR-206-depleted animals, we assessed neutrophil migration to a sterile tail fin wound as an example of a non-infectious inflammatory stimulus (Fig 3D). The number of neutrophils at the wound site did not significantly differ between scramble control (S3 Movie) and miR-206 knockdown embryos (S4 Movie), indicating the increased neutrophil response observed in trunk infections is *M. marinum* infection-dependent.

The effect of miR-206 knockdown on macrophage recruitment to infection was also assessed via static imaging following trunk infection with *M. marinum*. At both 1 and 3 dpi, there was no difference in the fluorescent area of infection-associated macrophages between control infected and miR-206 knockdown infected embryos (S5 Fig).

Further analysis of *M. marinum*-neutrophil interactions was performed to dissect the role of miR-206 in controlling infection. Following trunk infection, control embryos showed an increasing bacterial burden at a per lesion level from 1–15 hpi, whereas the bacterial burden in miR-206 knockdown embryos remained stable (Fig 4A–4C).

The quantity of neutrophils co-localising with the bacterial granuloma was then assessed to further confirm the increased interaction between neutrophils and *M. marinum* in miR-206 knockdown embryos. From 1–5 hpi, miR-206 had consistently higher neutrophil fluorescence colocalization with *M. marinum* fluorescence than control embryos (Fig 4D-4F). As with the neutrophil levels observed in the time-lapse analysis, at 10 hpi there was no difference in the numbers of neutrophils associating with bacteria, however the response in control infected embryos began to decline at 15 hpi while numbers were maintained in the miR-206 knockdown embryos.

Across the first 15–24 hours of infection, *M. marinum* was closely associated with more neutrophils in miR-206 knockdown embryos. The infection-associated increase in neutrophils and the higher levels of neutrophils localising at the granuloma implicate the increased neutrophil response as a deciding factor in the control of infection and prevention of bacterial spread.

## The Cxcr4/Cxcl12 signalling axis is downstream of miR-206 in *M. marinum* infection

To confirm the hypothesised link between the observed increased transcription of *cxcr4b* and *cxcl12a* and reduced bacterial burden through an increased neutrophil response early in infection, both genes were targeted for knockdown by CRISPR-Cas9. As both *cxcr4b* and *cxcl12* are involved in neutrophil migration and haematopoiesis, a reduction in their expression was expected to result in a reduced neutrophil response to infection and therefore an increased bacterial burden, reducing the protective effect of miR-206 knockdown.

Static imaging at 3 dpi revealed that double knockdown of *cxcr4b* and miR-206 ablated the increased neutrophil number associated with miR-206 knockdown (Fig 5A). Furthermore, addition of *cxcr4b* knockdown to miR-206 knockdown dampened the miR-206 knockdown-induced increase in neutrophil recruitment to a trunk infection and increased bacterial burden back to control levels (Fig 5B–5C). The effect observed in the double knockdown is consistent with a reduction in Cxcr4 and therefore the neutrophil response in infection via

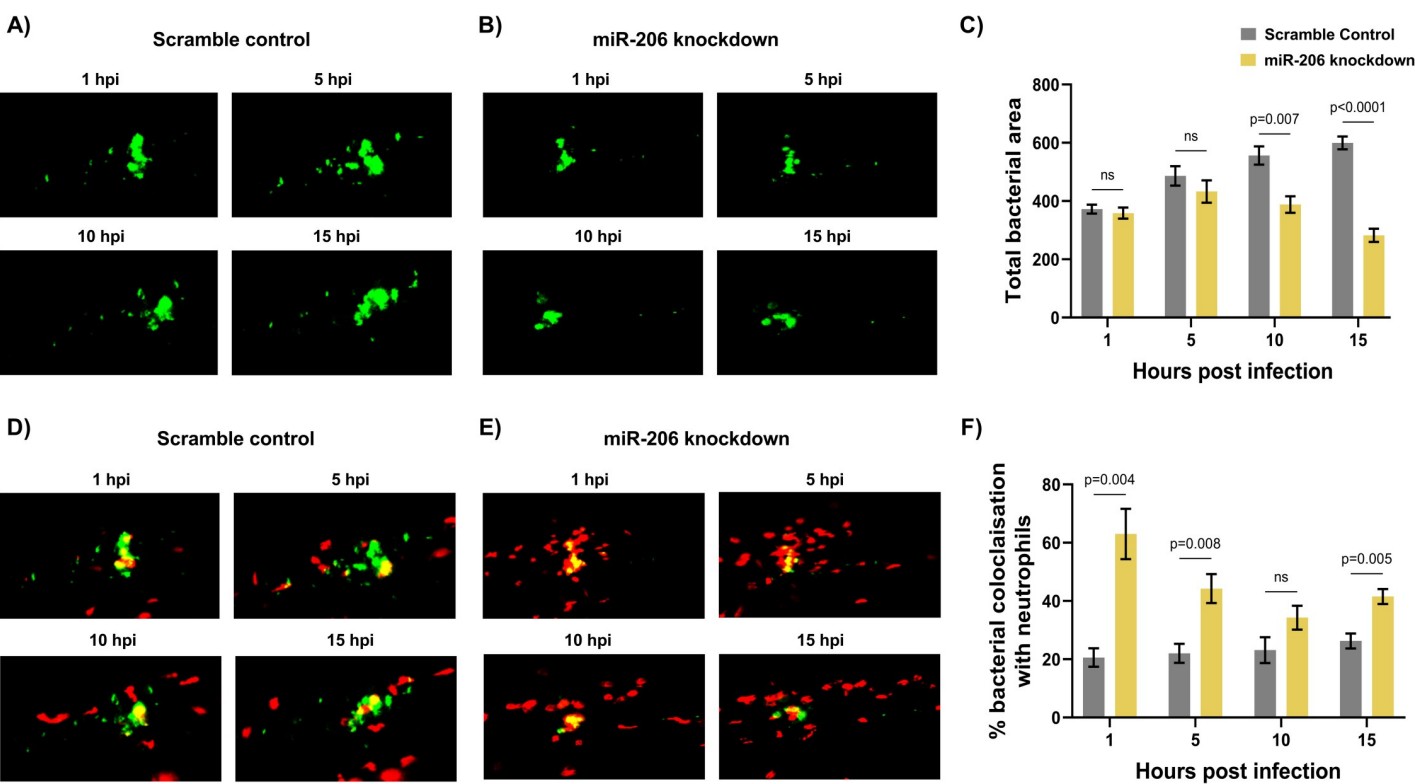

**Fig 4. Increased miR-206 knockdown-associated neutrophils prevent bacterial dissemination.** (A-B) Representative images of bacterial granulomas in trunk-infected control and miR-206 knockdown embryos. (C) Quantification of *M. marinum* burden in trunk-infected control and miR-206 knockdown embryos. (D-E) Representative images of *M. marinum*-neutrophil interactions in trunk-infected control and miR-206 knockdown embryos. Neutrophils are red *Tg(lyzC:dsred)* and *M. marinum* is green (wasabi); co-localisation is indicated by yellow fluorescence. (F) Quantification of the proportion of the bacterial fluorescence overlapping with neutrophil fluorescence (co-localisation yellow fluorescence in D-E) in trunk-infected control and miR-206 knockdown embryos. Each data point represents the mean of 6 foci of infection from 6 separate embryos with SEM shown. Differences between groups was calculated using multiple t-tests and the Holm-Sidak method.

haematopoiesis and chemoattraction. This suggests the miR-206 associated increase in *cxcr4b* is contributing to the enhanced neutrophil migration and reduced bacterial burden.

To further confirm involvement of Cxcr4 downstream of miR-206, the CXCR4 antagonist AMD3100 was used to pharmacologically block Cxcr4 signalling. AMD3100 treatment reduced the total neutrophil numbers in all treatment groups, and as expected, whole-body neutrophil counts were reduced in miR-206 knockdown embryos that were also treated with AMD3100 compared to miR-206 knockdown alone (Fig 5D). AMD3100 treatment decreased neutrophil recruitment to the site of infection compared to both control infected and miR-206 knockdown infected embryos (Fig 4E). Bacterial burden was increased in AMD3100 treated embryos and AMD3100 treatment of miR-206 knockdown embryos restored bacterial burden to control levels (Fig 5F).

Consistent with the results of *cxcr4b* knockdown, addition of *cxcl12a* knockdown to miR-206 knockdown decreased the total number of neutrophils and number of neutrophils recruited to sites of infection compared to miR-206 knockdown alone (Fig 6A and 6B), and increased the bacterial burden compared to miR-206 knockdown alone (Fig 6C).

## Discussion

In this study we have demonstrated an *in vivo* link between infection-induced miR-206 expression and the Cxcl12/Cxcr4 signalling axis in the control of mycobacterial infection.

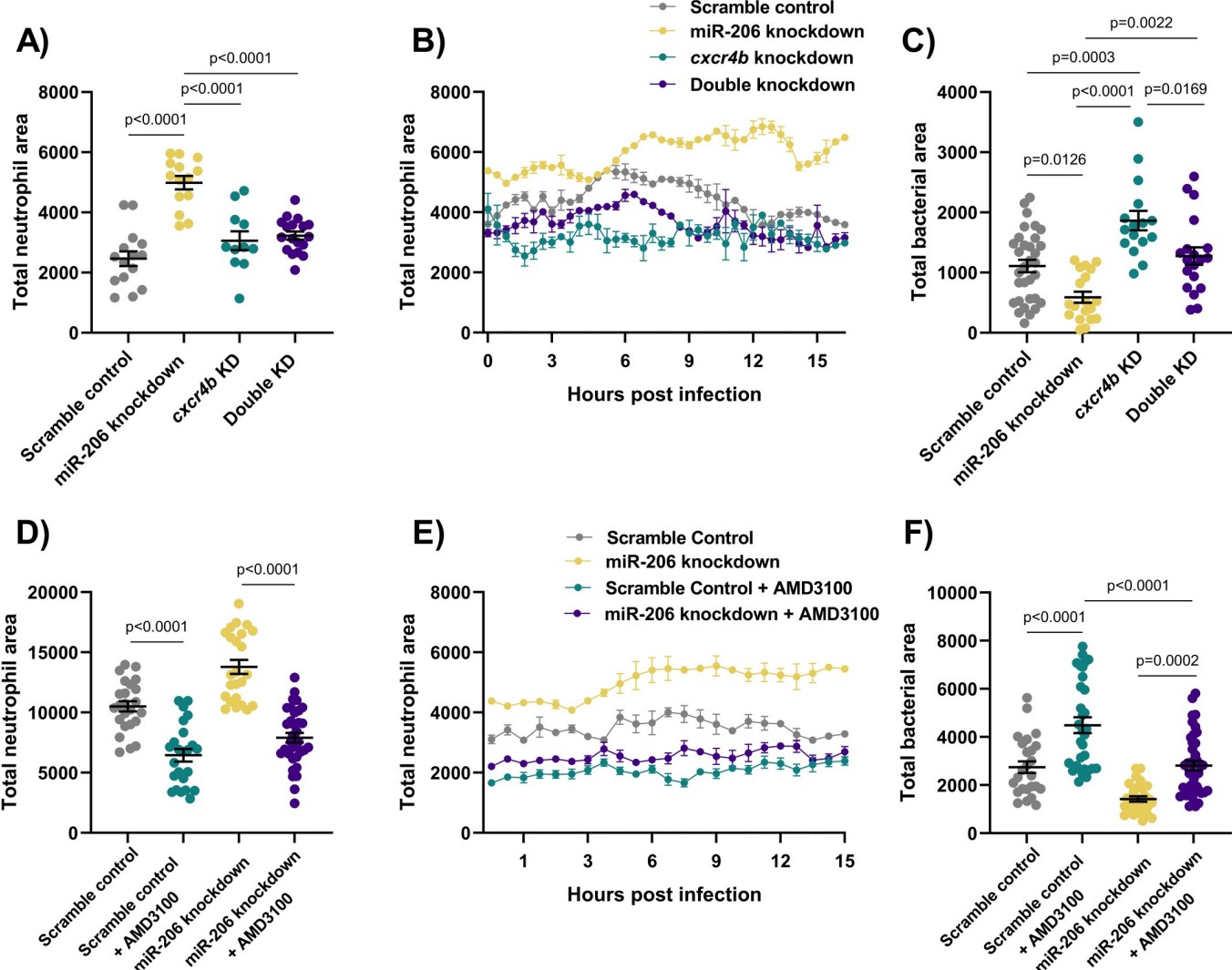

**Fig 5. Cxcr4 reduction places the Cxcl12/Cxcr4 signalling axis downstream of miR-206.** (A) Whole body neutrophil counts at 3 dpi of *cxcr4b* and double (*cxcr4b* and miR-206) knockdown embryos. (B) Measurement of neutrophil levels following trunk infection with *M. marinum* in double knockdown embryos. (C) Bacterial burden at 3 dpi in *M. marinum*-infected double knockdown embryos. (D) Whole body neutrophil counts at 3 dpi in miR-206 knockdown embryos treatment with AMD3100. (E) Measurement of neutrophil recruitment to *M. marinum* following trunk injection in miR-206 knockdown embryos treatment with AMD3100. (F) Bacterial burden at 3 dpi in miR-206 knockdown embryos treatment with AMD3100. Each data point represents a single measurement, with the mean and SEM shown. For time-lapse imaging, each data point represents the mean of 6 foci of infection from 6 separate embryos. Bacterial burden analysis was performed on 15–25 embryos per treatment. Graphs are representative of 2 biological replicates, except for AMD3100 data, which is a single biological replicate.

Knockdown of miR-206 resulted in a decreased bacterial burden and improved infection outcome. We attribute the reduced bacterial burden to an increased early neutrophil response from increased *cxcr4b* and *cxcl12a* transcript abundance in miR-206 knockdown animals. We show that host miR-206 is increased by pathogenic *M. marinum* to impede the host Cxcl12/Cxcr4 signalling axis, thereby reducing protective early neutrophil recruitment to the site of infection, aiding the creation of a permissive niche for mycobacterial infection (S6 Fig).

As this early protective neutrophil response was specific to virulent intracellular mycobacteria, the observed increase in miR-206 was deemed to be ESX1-dependent. We hypothesise that this may be a mycobacteria-driven response to avoid neutrophil phagocytosis and potentially oxidative killing [40–42]. Neutrophils are one of the first immune cells to respond to mycobacterial

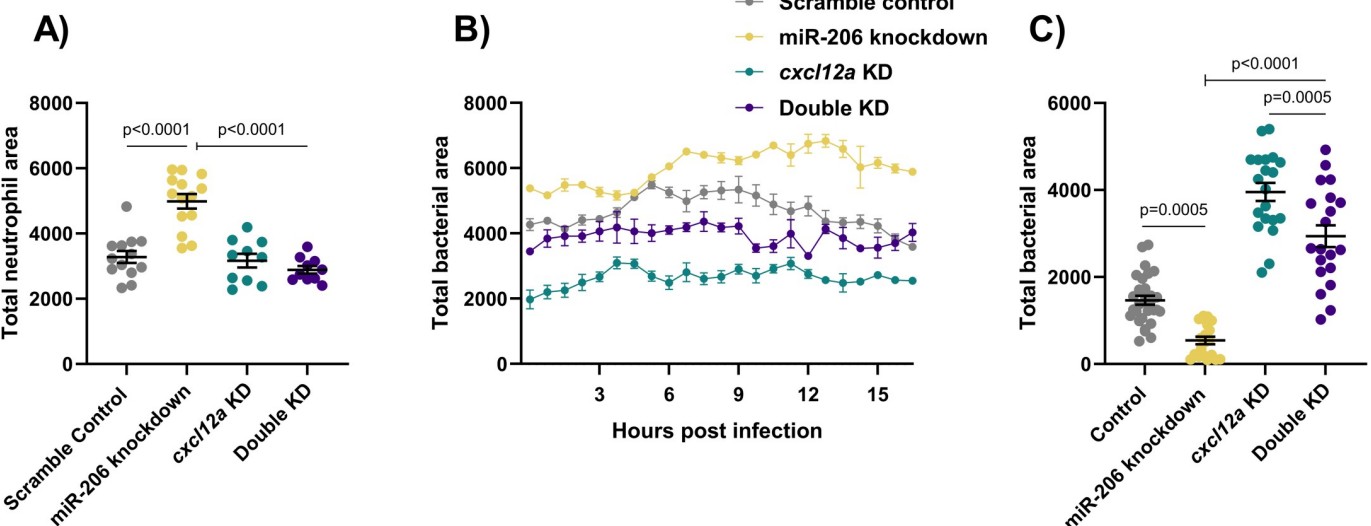

**Fig 6. Cxcl12a knockdown places the Cxcl12/Cxcr4 signalling axis downstream of miR-206.** (A) Whole body neutrophil counts at 3 dpi of *cxcl12a* and double (*cxcl12a* and miR-206) knockdown embryos. (B) Measurement of neutrophil recruitment to *M. marinum* following trunk injection in double knockdown embryos. (C) Bacterial burden at 3 dpi in *M. marinum*-infected double knockdown embryos. Each data point represents a single measurement, with the mean and SEM shown. For time-lapse imaging, each data point represents the mean of 4 foci of infection from 4 separate embryos. Bacterial burden analysis was performed on 20–30 embryos per treatment. Graphs are representative of 2 biological replicates.

infection and are capable of both phagocytosing and trapping mycobacteria in neutrophil extra-cellular traps [43]. Therefore, it is not surprising that mycobacteria have evolved a strategy to actively subvert early host neutrophil recruitment by reducing Cxcl12/Cxcr4 signalling, limiting the downstream exposure of mycobacteria to phagocytosis and oxidative killing.

Involvement of Cxcr4 and its ligand Cxcl12 in inflammation is well documented [44–46], however, previous studies have largely focused on their role in viral responses. Cxcr4 expression is reduced in the lymphocytes of leprosy patients but increased in *M. tuberculosis*-infected macrophages [47,48]. Recent work has highlighted Cxcr4 as a mediator of host infection-associated angiogenesis [49], while this study further links the Cxcl12/Cxcr4 signalling axis to virulence-dependent neutrophil recruitment during mycobacterial infections. Cxcr4 has also been shown to participate in pathogen immune evasion via interaction with TLR2, suggesting it may play an active role in other aspects of mycobacterial pathogenesis [50].

Our data provide precedence for cross-species conservation of host miRNA responses to mycobacterial infection. Although current understanding of the role of miR-206 in bacterial infections has been limited to date, previous investigation using *M. tuberculosis* infection of THP-1 cells revealed a similar infection-induced upregulation of miR-206 *in vitro* [15]. Our *in vivo* model of mycobacterial infection has allowed the interrogation of neutrophil responses as a downstream cellular response controlled by miR-206.

While we have demonstrated the Cxcr4/Cxcl12 pathway and neutrophil response as important in controlling *M. marinum* infection, it is likely that altered transcript levels of other profiled targets are concomitantly contributing to the effect of miR-206 on mycobacterial infection. One validated target of miR-206, VEGF, plays a significant role in the later development of granulomas during mycobacterial infection, consistent with the late downregulation of miR-206 that we observed [33,35,51]. Infection-induced VEGF signalling results in an aberrant angiogenesis programme which favours mycobacterial growth and spread [39,52–54]. This effect may be synergistic with Cxcl12/Cxcr4 signalling, which supports granuloma-associated angiogenesis through a Vegf-independent mechanism in zebrafish embryos [49].

Suppression of *elmo1* by miR-206 may further contribute to the immune avoidance associated with the infection-induced increase in miR-206. Recent investigations have revealed a role for ELMO1 in neutrophil migration and engulfment of apoptotic cells [55] and this has been linked to enhanced intracellular mycobacterial growth [56]. Increased transcription of Elmo1 following miR-206 knockdown is likely to increase neutrophil mobility during infection in cooperation with increased Cxcl12/Cxcr4 signalling.

miR-206 may additionally act on *TIMP3* to inhibit the activity of MMP9 during mycobacterial infection [15], preventing macrophage recruitment and granuloma formation in our miR-206 knockdown model [57]. However, dissecting the miR-206-MMP9 interaction may require a different experimental platform to determine if reduced *mmp9* expression is a result of transcriptional feedback from its inhibitor Timp3 or caused by the reduced bacterial burden. This may prove to be an additional pathway modulated by miR-206 during infection, acting to alter disease progression and highlights the complex interaction of miRNA and their multiple targets.

The final potential target gene we profiled, *MMD*, may also be of significance through the positive regulation of macrophage activation and downstream cytokine signalling cascades, however requires more investigation in mycobacterial pathogenesis as macrophage responses were not altered in miR-206 knockdown embryos [58].

It is evident that the selected target genes of miR-206 play crucial roles in the immune response and potentially in the outcome of pathogenic mycobacterial infections. We chose to assess the neutrophil responses as alterations in Cxcr4/Cxcl12 signalling would likely impact the mobility and responsiveness of neutrophils to pathogens. Recent studies on the role of the neutrophils in early infection suggest a host-protective response against mycobacteria [40,59], and we hypothesised that these cells may be responsible for the reduced disease burden in miR-206 knockdown embryos. It is, however, likely that the alteration of targets *Elmo1* and *Timp3* may be contributing to this effect and that role of these genes in mycobacterial infection must be further examined, especially in regard to their involvement in the miR-206/Cxcr4/Cxcl12 signalling axis.

Other pathways which interact with the Cxcr4/Cxcl12 axis and also regulate neutrophil responses may be adding to the increased neutrophil response. Chemokine pathways including Cxcl8, Cxcl18 Cxcr2 have been shown to be key mediators of neutrophil chemotaxis and immune responses [60–62]. Cxcr2 has previously been identified as regulating neutrophilic infiltration in mycobacterial infection and may further increase chemotaxis of neutrophils to sites of inflammation through Cxcl18b. A positive feedback system between CXCR2 and CXCR4 has been identified, suggesting that the miR-206 induced decrease in *cxcr4* in our system could also result in decreased *cxcr2*, compounding the suppression of neutrophil recruitment to infection [63]. This highlights the complexity of interconnected immune responses to mycobacteria and the need for further exploration of the regulation of these pathways.

In summary, we have identified potential target genes of miR-206 which may be biologically active during mycobacterial infection. We have demonstrated a link between infection-associated upregulation of miR-206 and suppression of neutrophil recruitment to the site of pathogenic mycobacterial infection involving the Cxcl12/Cxcr4 signalling pathway. This host response to infection by pathogenic mycobacteria appears to be conserved across host-pathogen pairings and could inform the development of biomarker or therapeutic strategies.

## Methods

### Ethics statement

Adult zebrafish were housed at the Centenary Institute and experiments were approved by Sydney Local Health District AWC Approval 17–036. The embryos were obtained by natural spawning and were raised in E3 media and maintained at 28–32˚C.

## Zebrafish lines

Zebrafish were AB strain. Transgenic lines used were: *Tg(lyzC:GFP)^{nz117}* and *Tg(lyzC:DsRed2)^{nz50}* were used for neutrophil imaging experiments [64].

## Embryo microinjection with antagomiR

Embryos were obtained by natural spawning and were injected with either miR-206 antago-miR (-CCACACACUUCCUUACAUUCCA-) or a scramble control (-CAGUACUUUUGU GUAGUACAA-) (GenePharma, China) at 200 pg/embryo at the single cell stage and maintained at 32°C.

## miRNA target prediction

Prediction of target mRNA was performed using TargetScan. dre-miR-206-3p was entered into TargetScanFish 6.2 (http://www.targetscan.org/fish_62/), hsa-miR-206 entered into TargetScan 7.2 (http://www.targetscan.org/vert_72/), and mmu-miR-206 entered into TargetScanMouse 7.2 (http://www.targetscan.org/mmu_72/).

## M. marinum culture

*M. marinum* M strain expressing Wasabi or tdTomato fluorescent protein was cultured and injected as previously described [65]. Briefly, *M. marinum* was grown at 28°C in 7H9 supplemented with OADC and 50 μg/mL hygromycin to an OD600 of approximately 0.6 before being washed and sheared by aspiration through a 32 G needle into single cell preparations that were then aliquoted and frozen in 7H9 at -80°C until needed. The concentration of bacteria was quantified from thawed aliquots by CFU recovery on 7H10 supplemented with OADC and 50 μg/mL hygromycin and grown at 28°C.

## UPEC culture

Uropathogenic *Escherichia coli* (UPEC) carrying the mCherry PGI6 plasmid was cultured in LB supplemented with 50 μg/mL of spectinomycin overnight at 37°C with 200 RPM shaking. Bacteria was then further diluted 1:10 with LB + spectinomycin (50 μg/ml) and incubated for 3 hours at 37°C with 200 RPM shaking. 1 mL of culture was centrifuged (16,000 x *g* for 1 minute), and the pellet washed in PBS. Following another centrifugation, the bacterial pellet was resuspended in 300 μl of PBS + 10% glycerol and aliquoted for storage. Enumeration of bacteria was performed by serial dilution on LB + spectinomycin agar plates and culturing at 37°C overnight. Bacterial concentration was determined by CFU counts.

## UPEC plasmid construction

The plasmid pGI6 was constructed by replacing the open reading frame (ORF) of msfGFP in pGI5 [66] with an E. coli codon-optimised ORF for mCherry. The mCherry ORF was first amplified with the forward primer (GCG CCG CCA TGG GTG AGC AAG GGC GAG GAG GAT) and reverse primer (GGC CCG GGA TCC TTA CTT GTA CAG CTC GTC CAT GCC) from the template pIDJL117 [67]. The PCR fragment was cloned at NcoI and BamHI in pGI5, thus replacing msfGFP, and the PCR-generated confirmed by sequencing.

## Bacterial infections

Staged at approximately 1.5 dpf, embryos were dechorionated and anesthetised in tricaine (160 μg/ml). Working solutions of *M. marinum* or UPEC (diluted with 0.5% w/v phenol red

**Table 1. Guide RNA sequences used for CRISPR-Cas9 mediated knockdown experiments.**

| Target | Primer |
|---|---|
| cxcr4b target 1 | TAATACGACTCACTATAGGAGCTCTGACTCCGGTTCTGTTTTAGAGCTAGAAATAGC |
| cxcr4b target 2 | TAATACGACTCACTATAGGACTGCAAGATAGCGGTCCGTTTTAGAGCTAGAAATAGC |
| cxcr4b target 3 | TAATACGACTCACTATAGGGTACCCATGCTCGAATTGGTTTTAGAGCTAGAAATAGC |
| cxcr4b target 4 | TAATACGACTCACTATAGGCTTACTGTGCCGGCATCCGTTTTAGAGCTAGAAATAGC |
| cxcl12a target 1 | TAATACGACTCACTATAGGTCGTAGTAGTCGCTCTGAGTTTTAGAGCTAGAAATAGC |
| cxcl12a target 2 | TAATACGACTCACTATAGGTCATGCACCGATTTCCAAGTTTTAGAGCTAGAAATAGC |
| cxcl12a target 3 | TAATACGACTCACTATAGGATACTCACATGACTTGGAGTTTTAGAGCTAGAAATAGC |
| cxcl12a target 4 | TAATACGACTCACTATAGGGCAGATACTCACATGACTGTTTTAGAGCTAGAAATAGC |
| scramble target 1 | TAATACGACTCACTATAGGCAGGCAAAGAATCCCTGCCGTTTTAGAGCTAGAAATAGC |
| scramble target 2 | TAATACGACTCACTATAGGTACAGTGGACCTCGGTGTCGTTTTAGAGCTAGAAATAGC |
| scramble target 3 | TAATACGACTCACTATAGGCTTCATACAATAGACGATGGTTTTAGAGCTAGAAATAGC |
| scramble target 4 | TAATACGACTCACTATAGGTCGTTTTGCAGTAGGATCGGTTTAGAGCTAGAAATAGC |

dye) were injected into either the caudal vein or trunk to deliver approximately 200 CFU *M. marinum* or 250 CFU UPEC. Embryos were recovered in E3 media + PTU (0.036 g/L) and housed at 28˚C.

## CRISPR-Cas9 mediated knockdown

Embryos were injected at the 1–2 cell stage with 1 nL of CRISPR mixture containing 1 μg/μl Guide (g) RNA (Table 1), 500 μg/mL Cas9. For double knockdowns with CRISPR-Cas9 and antagomiR, mixtures contained 1 μg/μl gRNA, 100 pg/nL antagomiR (miR-206), and 500 μg/mL Cas9. gRNA was synthesised as previously described [68]. Embryos were transferred to E3 containing methylene blue and maintained at 32˚C.

## Gene expression analysis

Groups of 10 embryos were lysed and homogenised using a 27-gauge needle in 500 μl Trizol (Invitrogen) and RNA extracted as per the manufacturer's instructions. cDNA was synthesised from 500 ng RNA using the miScript II RT kit (Qiagen) with HiFlex buffer. qPCR was carried out on an Mx3000p Real-time PCR system using Quantitect SYBR Green PCR Mastermix and primer concentration of 300 nM (Table 2). For miRNA qPCRs, the miScript Universal Primer was used alongside miR specific miScript primer assays (miR-206 cat. no. MS00001869 and U6 cat. no. MS00033740; Qiagen).

Cycling conditions for miRNA were: 95˚C for 15 minutes; 40 cycles of 95˚C for 20 seconds, 56˚C for 30 seconds, 72˚C for 30 seconds with fluorescence data acquisition occurring at the end of each cycle, followed by 1 cycle of 95˚C for 1 minute, 65˚C for 30 seconds, and 97˚C for 30 seconds. For mRNA, the conditions were: 95˚C for 15 minutes; 40 cycles of 94˚C for 15 seconds, 55˚C for 30 seconds, 70˚C for 30 seconds with fluorescence data acquisition occurring at the end of each cycle, followed by 1 cycle of 95˚C for 1 minute, 65˚C for 30 seconds, and 97˚C for 30 seconds.

U6 or β-actin was used as an endogenous control for normalisation and data analysed using the $2^{-\Delta\Delta}$ Ct method.

## AMD3100 treatment

Embryos were treated with 20 μM AMD3100 (Sigma-Aldrich), a pharmacological CXCR4 antagonist, dissolved in water and refreshed daily.

**Table 2. qPCR primer sequences.**

| qPCR Primer | Sequence 5'-3' | Ensembl ID |
|---|---|---|
| cxcr4a forward | CAGTTTGGACCGGTACCTCG | ENSDARG00000057633 |
| cxcr4a reverse | CCAGGTGACAAACGAGTCCT | |
| cxcr4b forward | TCGCAGACCTCCTGTTTGTC | ENSDARG00000041959 |
| cxcr4b reverse | CCTTCCCGCAAGCAATTTCC | |
| cxcl12a forward | ATTCGCGAGCTCAAGTTCCT | ENSDARG00000037116 |
| cxcl12a reverse | ATATCTGTGACGGTGGGCTG | |
| elmo1 forward | TGTTGACCATGCGTCTCAGT | ENSDARG00000098753 |
| elmo1 reverse | CCACCTTCACGATGTCTGCC | |
| mmd forward | GGGGGTCTGGTCTACTGTCT | ENSDARG00000040387 |
| mmd reverse | TTGTTAGTGGCTCAGGCGTC | |
| vegfaa forward | TCCCGACAGAGACACGAAAC | ENSDARG00000045971 |
| vegfaa reverse | TTTACAGGTGAGGGGGTCCT | |
| b-actin forward | CCTTCCAGCAGATGTGGATT | ENSDARG00000037870 |
| b-actin reverse | CACCTTCACCGTTCCAGTTT | |

## Static imaging and burden analyses

Live imaging was performed on anaesthetised embryos on a depression microscope slide. Images were acquired using a Leica M205FA Fluorescent Stereo Microscope equipped with a Leica DFC365FX monochrome digital camera (Leica Microsystems, Germany). Images were analysed using ImageJ software to quantify the fluorescent pixel count, defined as fluorescent signal above a consistent set background determined empirically for each experimental dataset [65]. Data is presented as total fluorescent area (pixels) above background level.

## Neutrophil tracking analyses

Time-lapse imaging was performed on a Deltavision Elite at 28°C (GE, USA). Following infection with *M. marinum* into the trunk, embryos were mounted in a 96-well black-walled microplate in 1% low-melting point agarose topped up with E3. Images were captured every 60–180 seconds for 16–24 hours. Analysis was performed using ImageJ software. Briefly, images were analysed for the quantity of neutrophil fluorescence in a 1000 x 500 μm box around infection foci by quantifying the fluorescent pixel count (total neutrophil area) at each time point. Tracking of individual neutrophil retention at the site of infection was performed manually using analysis of timestamped images.

## Statistics

Statistical analysis was performed in GraphPad Prism (v. 9.0.0). All data was analysed by T-test or ANOVA depending on the number of experimental groups, post-hoc analysis performed using Tukey's multiple comparisons test. For time-lapse data, group comparisons were computed using the Sidak test. Outliers were removed prior to statistical analysis using ROUT, with Q = 1%.

## Supporting information

**S1 Fig. miR-206 antagomiR knockdown at 5 dpi.** Expression of miR-206 was analysed by qPCR at 5 dpi following antagomiR knockdown. Each data point represents a single measurement of 10 pooled embryos and 2 biological replicates, with the mean and SEM shown. (TIF)

**S2 Fig. Predicted *cxcr4a*, *cxcr4b*, and *cxcl12a* 3'UTR binding sites to miR-206 seed sequence.** Alignment of potential target gene binding sites to miR-206 seed sequence predicted by TargetScan. (TIF)

**S3 Fig. Expression of *cxcr4b* and *cxcl12a* during *M. marinum* infection.** *cxcr4b* and *cxcl12a* transcript abundance was measured by qPCR in *M. marinum* infected embryos at 1, 3 and 5 dpi by qPCR. Each data point represents 10 embryos and contains 2 biological replicates. (TIFF)

**S4 Fig. Effect of *cxcr4b* and *cxcl12a* expression on neutrophil responses throughout infection.** Whole-body neutrophil fluorescence at 1, 3, and 5 dpi in control and miR-206 knockdown embryos. Each data point represents a single neutrophil with the mean and SEM shown. (TIFF)

**S5 Fig. Granuloma associated macrophages in miR-206 knockdown embryos.** Following trunk infection with *M. marinum*, total macrophage fluorescence was measured at sites of infection. Each data point represents a single embryo with the mean and SEM shown. (TIFF)

**S6 Fig. Summary of findings.** Top line: *M. marinum* infection induces the expression of host miR-206. miR-206 suppresses the expression of *cxcr4b* and *cxcl12a* which results in suboptimal neutrophil recruitment and supports bacterial growth. Bottom line: AntagmiR-mediated neutralisation of miR-206 allows increased expression of *cxcr4b* and *cxcl12a* which increases neutrophil recruitment and suppresses bacterial growth. (TIF)

**S1 Table. miR-206 target prediction results from TargetScan analysis.** TargetScan output of predicted mRNA targets of zebrafish, mouse, and human miR-206. (XLSX)

**S1 Data. Source data for graphs.** (XLSX)

**S1 Movie. Neutrophil migration to infection in control embryos.** Neutrophils are red (*lyzC*: *dsred*) and *M. marinum* is green (wasabi); co-localisation is indicated by yellow fluorescence. (AVI)

**S2 Movie. Neutrophil migration to infection in miR-206 knockdown embryos.** Neutrophils are red (*lyzC*:*dsred*) and *M. marinum* is green (wasabi); co-localisation is indicated by yellow fluorescence. (AVI)

**S3 Movie. Neutrophil migration to sterile wound site in control embryos.** Neutrophils are green (*lyzC*:*GFP*). (AVI)

**S4 Movie. Neutrophil migration to sterile wound site in miR-206 knockdown embryos.** Neutrophils are green (*lyzC*:*GFP*). (AVI)

## Acknowledgments

Dr Angela Kurz of the BioImaging Facility and Sydney Cytometry at Centenary Institute for technical assistance with imaging. Dr Elinor Hortle for assistance with image and data analysis, and Dr Pradeep Manuneedhi Cholan for assistance with imaging.

## Author Contributions

**Conceptualization:** Kathryn Wright, Kumudika de Silva, Karren M. Plain, Auriol C. Purdie, Warwick J. Britton, Stefan H. Oehlers.

**Data curation:** Kathryn Wright, Stefan H. Oehlers.

**Formal analysis:** Kathryn Wright, Stefan H. Oehlers.

**Funding acquisition:** Kumudika de Silva, Karren M. Plain, Auriol C. Purdie, Warwick J. Britton, Stefan H. Oehlers.

**Investigation:** Kathryn Wright.

**Methodology:** Kathryn Wright, Stefan H. Oehlers.

**Resources:** Tamika A. Blair, Iain G. Duggin.

**Supervision:** Warwick J. Britton, Stefan H. Oehlers.

**Writing – original draft:** Kathryn Wright, Stefan H. Oehlers.

**Writing – review & editing:** Kathryn Wright, Kumudika de Silva, Karren M. Plain, Auriol C. Purdie, Warwick J. Britton, Stefan H. Oehlers.

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
