## [Decision Letter · Decision Letter 0]

2 Feb 2021

Dear Dr. Oehlers,

Thank you very much for submitting your manuscript "Mycobacterial infection-induced miR-206 inhibits protective neutrophil recruitment via the CXCL12/CXCR4 signalling axis" for consideration at PLOS Pathogens. As with all papers reviewed by the journal, your manuscript was reviewed by members of the editorial board and by several independent reviewers. In light of the reviews (below this email), we would like to invite the resubmission of a significantly-revised version that takes into account the reviewers' comments. In particular the effects of miR-206 on killing of bacteria is important.

We cannot make any decision about publication until we have seen the revised manuscript and your response to the reviewers' comments. Your revised manuscript is also likely to be sent to reviewers for further evaluation.

Sincerely,

JoAnne L Flynn, PhD

Section Editor

PLOS Pathogens

JoAnne Flynn

Section Editor

PLOS Pathogens

Kasturi Haldar

Editor-in-Chief

PLOS Pathogens

orcid.org/0000-0001-5065-158X

Michael Malim

Editor-in-Chief

PLOS Pathogens

orcid.org/0000-0002-7699-2064

Reviewer's Responses to Questions

**Part I - Summary**

Reviewer #1: In this manuscript, Kathryn Wright et al. have used the zebrafish model of Mycobacterium marinum (M.m) infection and found that M.m induces a miRNA, miR-206 during infection of the embryos. They investigated the role of miR206 in the early stages of granuloma formation by challenging zebrafish embryos with the bacterium. The authors found that miR-206 upregulation coincides with more bacterial foci and knocking down miR206 with antagomir- mediated knockdown decreased bacterial burden. To identify the host targets of miR-206, they checked the expression of a chemokine and chemokine receptor (by using a target prediction algorithm, though the results of this analysis was not presented!) and demonstrated that miR-206 knock down leads to upregulation of Cxcr4a/b, elmo1, Cxcl12 during M.m infection. Upregulation of the CXCR4/CXCL12 genes following miR-206 knock down resulted in an increased neutrophil response at the granulomatous foci and associated with reduction in bacterial foci. Wright et al. then used two different strategies (gene knock down by CrispR/Cas9 and pharmacological inhibition of Cxcr4 pathway with the selective inhibitor Plerixafor/AMD3100 to show that inhibiting the CXCR4/CXCL2 signaling pathway blunts the neutrophil response at the infection site, that ultimately result in high bacterial burden. Based on these results, the authors propose that M.m infection induces miR-206 expression to prevent a protective neutrophil response early during granuloma formation that might lead to a better outcome. The manuscript is well written, and experiments are well controlled. The identification of miR-206 mediated targeting of cxcr4/cxcl12 signaling and downstream neutrophil recruitment adds to the current understanding of protective innate immunity to mycobacterial infections.

Reviewer #2: This manuscript by Wright et al investigates the role of miR-206, a microRNA the authors note is associated with myoblast differentiation and muscle development, and neutrophil responses in Mycobacterium marinum infection. The authors use the zebrafish model for their work and find that miR-206 inhibits the CXCL12/CXCR4 pathway to inhibit neutrophil infiltration to early granulomas and this leads to higher bacteria loads per animal. Neutrophilic inflammation is often associated with bad outcomes in mycobacterial infections but there is some data suggesting they play important protective roles in acute infection and the author’s data supports this contention. The zebrafish model is likely to be the best system to investigate this stage of infection and the author’s use of it to address this mechanistic question is novel. The overall findings are interesting and add some new information to the field that will be of interest to researchers working on innate immunity in TB.

Reviewer #3: The manuscript informative, mechanistic, well-written study on a timely TB topic (miRNA and neutrophil recruitment to granulomas) that will be of interest to the audience. The work highlights many of the best qualities about the zebrafish/marinum model.

**Part II – Major Issues: Key Experiments Required for Acceptance**

Reviewer #1: In this manuscript, Kathryn Wright et al. have used the zebrafish model of Mycobacterium marinum (M.m) infection and found that M.m induces a miRNA, miR-206 during infection of the embryos. They investigated the role of miR206 in the early stages of granuloma formation by challenging zebrafish embryos with the bacterium. The authors found that miR-206 upregulation coincides with more bacterial foci and knocking down miR206 with antagomir- mediated knockdown decreased bacterial burden. To identify the host targets of miR-206, they checked the expression of a chemokine and chemokine receptor (by using a target prediction algorithm, though the results of this analysis was not presented!) and demonstrated that miR-206 knock down leads to upregulation of Cxcr4a/b, elmo1, Cxcl12 during M.m infection. Upregulation of the CXCR4/CXCL12 genes following miR-206 knock down resulted in an increased neutrophil response at the granulomatous foci and associated with reduction in bacterial foci. Wright et al. then used two different strategies (gene knock down by CrispR/Cas9 and pharmacological inhibition of Cxcr4 pathway with the selective inhibitor Plerixafor/AMD3100 to show that inhibiting the CXCR4/CXCL2 signaling pathway blunts the neutrophil response at the infection site, that ultimately result in high bacterial burden. Based on these results, the authors propose that M.m infection induces miR-206 expression to prevent a protective neutrophil response early during granuloma formation that might lead to a better outcome. The manuscript is well written, and experiments are well controlled. The identification of miR-206 mediated targeting of cxcr4/cxcl12 signaling and downstream neutrophil recruitment adds to the current understanding of protective innate immunity to mycobacterial infections. However, few concerns remain that needs the author’s attention.

1. Whether an augmented neutrophil influx after miR-206 knock down directly causing the bacterial killing was not investigated.

2. The authors did not show the effect of miR-206 knock down on other immune cells especially macrophages and monocytes that may play a major antimicrobial role.

3. Since neutrophils were shown to be pathological during TB in multiple models of murine TB and humans, does the excessive infiltration of neutrophils early after miR-206 knock down affect granuloma fate/outcome at later time points? The neutrophil response was investigated up to 3dpi.

4. Since miR-206 expression declines after 3 dpi, what happens to the expression level of CXCR4 and CXCL12 ? This data would add value to the existing results. What is the effect of declining miR-206 expression on neutrophil response and bacterial burden at 5dpi? These data are important in making the conclusion the authors have made.

5. CXCR4/CXCL12 signaling is essential in retaining neutrophils in the bone marrow of mammals and perturbation of this signaling is a requisite for neutrophil mobilization. The high neutrophil response in the cxcr4 and cxcl12 kd zebrafish embryos suggest that the neutrophils are retained at the site of infection. Are these neutrophils newly recruited cells or ageing cells that are retained at the site of infection? Moreover, CXCR2 has been shown to regulate neutrophil influx during mycobacterial infection (Dorhoi etal, 2013; Lovewell et al. 2020). Was CXCR2 expression checked in this study or filtered by the algorithm? The authors need to comment on this.

Minor Comments:

1. Fig.1a. The fold changes should be expressed as relative to uninfected.

2. Fig. 1f. the x-axis should read as 6h and 1dpi as mentioned in the text and figure legend.

3. The authors must comment on the results of the predicted gene targets of miR-206 as checked in the figure 2 though the authors cited the original publications describing this bioinformatic target prediction algorithms (line 138)

4. The miR-206 and cxcr4b or cxcl12a DKD embryos had significantly fewer bacterial areas than Cxcr4b or cxcl12a KD embryos suggesting that other pathways might regulate bacterial control at granuloma foci. The authors suggested in addition to cxcr4/cxcl12 signaling, other pathways might be involved. The authors should discuss these putative pathways (though they mention about elmo1 which regulates cell motility) that are targets of miR-206.

Reviewer #2: The writing in the paper is generally clear and concise, but the authors do not provide enough background in the introduction to fully appreciate their work and the discussion could use a little more depth. More detail is needed on the zebrafish model, mRNAs in TB/mycobacterial infections, etc. The materials and methods section needs more detail than is currently provided in several areas so the reader doesn’t have to go back to other papers to find out important details (e.g. how bacteria were cultured and quantified, etc). This may also help with questions I had but weren’t addressed in the paper (e.g. why were the timepoints selected, why were neutrophils targeted, etc). It was also unclear why the CXCR4/CXCL12 pathway was selected in their analysis rather than the other genes they profiled (Fig. 2). These genes are briefly mentioned these other genes in the discussion but are not well integrated well into the story the authors are telling. The paper might also benefit from having a model showing how the author’s link miR-206, downregulated CXCR4/CXCL12, and decreased neutrophil infiltration with reduced bacterial control.

Reviewer #3: No new experiments requested.

**Part III – Minor Issues: Editorial and Data Presentation Modifications**

Reviewer #1: Minor Comments:

1. Fig.1a. The fold changes should be expressed as relative to uninfected.

2. Fig. 1f. the x-axis should read as 6h and 1dpi as mentioned in the text and figure legend.

3. The authors must comment on the results of the predicted gene targets of miR-206 as checked in the figure 2 though the authors cited the original publications describing this bioinformatic target prediction algorithms (line 138)

4. The miR-206 and cxcr4b or cxcl12a DKD embryos had significantly fewer bacterial areas than Cxcr4b or cxcl12a KD embryos suggesting that other pathways might regulate bacterial control at granuloma foci. The authors suggested in addition to cxcr4/cxcl12 signaling, other pathways might be involved. The authors should discuss these putative pathways (though they mention about elmo1 which regulates cell motility) that are targets of miR-206.

Reviewer #2: - Protein nomenclature for CXCR4/CXCL12/VEGF/MMP9/TIMP3 should be capitalized.

- CRISPR should be capitalized.

- ‘MicroRNA’ line 42 and 56 should be capitalized.

- ‘gene silencing’ doesn’t need quotation marks

- THP-1s would be better described as a monocyte-like cell line rather than as leukocytic cells

Reviewer #3: None

PLOS authors have the option to publish the peer review history of their article (what does this mean?). If published, this will include your full peer review and any attached files.

Reviewer #1: No

Reviewer #2: No

Reviewer #3: No
---

## [Decision Letter · Decision Letter 1]

29 Mar 2021

Dear Dr. Oehlers,

We are pleased to inform you that your manuscript 'Mycobacterial infection-induced miR-206 inhibits protective neutrophil recruitment via the CXCL12/CXCR4 signalling axis' has been provisionally accepted for publication in PLOS Pathogens.

Best regards,

Padmini Salgame

Associate Editor

PLOS Pathogens

JoAnne Flynn

Section Editor

PLOS Pathogens

Kasturi Haldar

Editor-in-Chief

PLOS Pathogens

orcid.org/0000-0001-5065-158X

Michael Malim

Editor-in-Chief

PLOS Pathogens

orcid.org/0000-0002-7699-2064

Reviewer Comments (if any, and for reference):

Reviewer's Responses to Questions

**Part I - Summary**

Reviewer #1: (No Response)

Reviewer #2: This manuscript is a revision of a previously submitted manuscript investigating the role of miR-206, a microRNA that regulates the chemokines CXCL12a and CXCL4, in Mycobacterium marinum infection. The authors used Zebrafish infected with this pathogen as a model for M. tuberculosis infection and investigated how miR-206 influenced neutrophil trafficking and antimicrobial activity in early stages of infection. Through a series of experiments involving fluorescent mycobacteria, deletion mutants, and antagomir-silenced gene expression, the authors found that infection induced miR-206 expression and this reduced CXCL12/CXCR4 signaling and neutrophil recruitment, leading to reduced control in acute infection. This is an interesting outcome, and this is probably the best model for investigating this topic. This paper will be interesting to people working in innate immunity in TB and neutrophil immunologists.

Many of my prior criticisms were centered on the manuscript construction (missing details, lack of background, etc), and the authors have addressed my concerns satisfactorily.

**Part II – Major Issues: Key Experiments Required for Acceptance**

Reviewer #1: (No Response)

Reviewer #2: No major concerns in this revision. My previously-indicated concerns have been adequately addressed.

**Part III – Minor Issues: Editorial and Data Presentation Modifications**

Reviewer #1: (No Response)

Reviewer #2: No concerns noted.

PLOS authors have the option to publish the peer review history of their article (what does this mean?). If published, this will include your full peer review and any attached files.

Reviewer #1: No

Reviewer #2: No

---

## [Editor Report · Acceptance letter]

4 Apr 2021

Dear Dr. Oehlers,

We are delighted to inform you that your manuscript, "Mycobacterial infection-induced miR-206 inhibits protective neutrophil recruitment via the CXCL12/CXCR4 signalling axis," has been formally accepted for publication in PLOS Pathogens.

Best regards,

Kasturi Haldar

Editor-in-Chief

PLOS Pathogens

orcid.org/0000-0001-5065-158X

Michael Malim

Editor-in-Chief

PLOS Pathogens

orcid.org/0000-0002-7699-2064